# Investigation of Structural Features of Two Related Lipases and the Impact on Fatty Acid Specificity in Vegetable Fats

**DOI:** 10.3390/ijms23137072

**Published:** 2022-06-25

**Authors:** Zehui Dong, Kim Olofsson, Javier A. Linares-Pastén, Eva Nordberg Karlsson

**Affiliations:** 1Biotechnology, Department of Chemistry, Lund University, 221 00 Lund, Sweden; javier.linares_pasten@biotek.lu.se (J.A.L.-P.); eva.nordberg_karlsson@biotek.lu.se (E.N.K.); 2AAK AB, Skrivaregatan 9, 215 32 Malmö, Sweden; kim.olofsson@aak.com

**Keywords:** substrate specificity, lipase, transesterification, fatty acid, 3D structure, substrate docking, molecular dynamics

## Abstract

One of the indispensable applications of lipases in modification of oils and fats is the possibility to tailor the fatty acid content of triacylglycerols (TAGs), to meet specific requirements from various applications in food, nutrition, and cosmetic industries. Oleic acid (C18:1) and stearic acid (C18:0) are two common long fatty acids in the side chain of triglycerides in plant fats and oils that have similar chemical composition and structures, except for an unsaturated bond between C9 and C10 in oleic acid. Two lipases from *Rhizomucor miehei* (RML) and *Rhizopus oryzae* (ROL)*,* show activity in reactions involving oleate and stearate, and share high sequence and structural identity. In this research, the preference for one of these two similar fatty acid side chains was investigated for the two lipases and was related to the respective enzyme structure. From transesterification reactions with 1:1 (molar ratio) mixed ethyl stearate (ES) and ethyl oleate (EO), both RML and ROL showed a higher activity towards EO than ES, but RML showed around 10% higher preference for ES compared with ROL. In silico results showed that stearate has a less stable interaction with the substrate binding crevice in both RML and ROL and higher tendency to freely move out of the substrate binding region, compared with oleate whose structure is more rigid due to the existence of the double bond. However, Trp88 from RML which is an Ala at the identical position in ROL shows a significant stabilization effect in the substrate interaction in RML, especially with stearate as a ligand.

## 1. Introduction

Lipases (EC 3.1.1.3), are the most used enzymes to modify the structure of oils and fats. They can be used to tailor the fatty acid content of natural lipids to meet special needs for use in, e.g., food, nutrition, and cosmetic applications. Lipases primarily hydrolyze ester bonds and can catalyze different reactions including, e.g., hydrolysis, acidolysis, esterification, transesterification, and interesterification reactions depending on substrate and reaction conditions [1].

Lipases are widely found in almost all kinds of living organisms, but mostly in animals, plants, and microbes [2]. Most of the research on lipase applications relies on commercially available lipases. Over 50% originate from and are produced by, microorganisms due to their higher robustness, activity varieties, and yield. Another advantage of microbial lipases is the ability to function without co-factors, whereas some animal lipases require coenzymes under some conditions [2,3,4].

Structurally lipases belong to the α/β hydrolase superfamily which also contains proteases, peroxidases, esterases, epoxide hydrolases, and dehalogenases. The enzymes included in this superfamily have a rather diverse function and share a rather low sequence identity. However, they all hold a conserved α/β-sheet core consisting in eight β strands connected by α helices which indicates a common ancestor origin [5]. Another feature of α/β hydrolases is the catalytic center formed by a conserved triad of Ser-Glu (Gln)-His as catalytic residues together with an oxyanion hole [6]. Due to the broad substrate specificity and significant regiospecificity, lipases are used for many industrial applications. Lipase substrate specificity has been a field of research attracting a lot of interest [7,8,9,10], as well as research focused on the chain length selectivity [11]. Compared with esterases, lipases show specifically high activity towards medium to long-chain water-insoluble triglycerides at a two-phase interface [12].

To categorize the many different types of lipases, a systematic database named Lipase Engineering Database (LED) has been developed (http://www.led.uni-stuttgart.de/, accessed on 15 May 2021), in which sequences of lipases, as well as some other homologous serine hydrolases were assigned into 112 homologous families and 38 superfamilies. One of the superfamilies identified in the database is the lipases from filamentous fungi (abH23), which has been a highly interesting lipase group that contains several commonly used lipases in oil processing, such as the lipases from *Rhizopus oryzae* (ROL), *Rhizomucor miehei* (RML) and *Thermomyces lanuginosus* (TLL) [1,2,13,14].

A flexible “lid” structure commonly exists in most lipases covering the active site, which is structurally closed in the pure aqueous environment and open in the presence of a hydrophobic surface. Lipases are inactive when the lid is closed. When the lid is open, the catalytic center gets exposed to substrates and lipases get activated. The lipases from filamentous fungi have this typical lid structure. Due to the fact that most lipases have this flexible lid domain located very close to the catalytic center, engineering in the lid domain has been shown to affect lipase activity, substrate specificity, thermostability, as well as stability towards detergents and organic solvents [15].

In this work, lipases from the two filamentous fungi: *Rhizopus oryzae* (ROL) and *Rhizomucor miehei* (RML), were selected in a study on the relationship between structure and function aiming at identifying structural features of importance for the selectivity for different types of fatty acid side chains, using stearic and oleic acid as a model. Oleic acid (C18:1) and stearic acid (C18:0) are two of the predominant fatty acids in cocoa butter [16,17]. The specific alteration between oleate and stearate at different positions on the triglyceride backbone is highly essential in chocolate applications since different triglycerides have different crystallization natures. Therefore, the lipases specificity for oleate and stearate substrate, respectively, can be an important parameter in the production of structured triglycerides [14].

ROL and RML have both been characterized with crevice binding types, which have been reported to contain three important substrate-binding regions: (1) the catalytic serine; (2) a hydrophobic, long, and deep binding crevice which has a 6 Å depth from the protein surface; (3) a hydrophobic, short, and shallow binding dent which is in parallel with the binding crevice [18]. ROL and RML share a relatively high sequence and structure similarity and show a preference for the middle to long fatty acid side chains. Oleic acid and stearic acid were chosen as fatty acids in transesterification reactions with 1-propanol to compare the substrate specificity of ROL and RML towards ethyl esters. In silico docking and molecular dynamic (MD) simulations were performed to predict which features that possibly affect the substrate preference, despite the small structural differences between both the two lipases (ROL and RML) and the two fatty acids (oleic acid and stearic acid).

## 2. Results and Discussion

### 2.1. Lipases Activity Profile

The activities of the lipases ROL and RML towards ethyl oleate (EO) and ethyl stearate (ES), respectively, were evaluated in transesterification reactions with a 1:1 molar ratio of EO:ES mixed with 1-propanol at 60 °C, 900 rpm (Figure 1). The two esters were mixed to: (i) allow evaluation of the substrate preference in a competing environment, and (ii) reduce the influence of any type of error based on sampling different reactions. The activities towards ES and EO were named V_ES_ and V_EO_, respectively.

The substrate preference of the two lipases was expressed as an activity ratio: V_ES_/V_EO_. The activity was calculated as the initial production rate of propyl stearate (PS) and propyl oleate (PO) (from ES and EO) in the transesterification reactions, with either of the two lipases. The concentration of PS and PO in the samples was plotted versus time to draw the reaction progress curve and the initial velocity was calculated as the slope of the linear region of the progress curve. (Plots are shown in Appendix A). V_ES_ and V_EO_ of both lipases with different enzyme dosages, together with the relative activity (V_ES_/V_EO_), are listed in Table 1. With each dosage of lipase, the reaction was repeated in triplicate.

As shown in Table 1, both ROL and RML show higher activity towards EO than towards ES despite variable loadings of lipases. However, ROL shows a higher preference towards EO by showing an average activity ratio V_ES_/V_EO_ of 0.62, compared with RML which shows the average ratio of 0.75. To further validate this difference with a statistical method, a 2-sample *t*-test was run, with the data of the activity ratio V_ES_/V_EO_ from ROL and from RML (Table 1), respectively as 2 samples. The *t*-test was run with Minitab (Minitab^®^ 19, LLC, State College, PA, USA). The result of the *t*-test was *p*-value = 0.00 < 0.05, which showed that it is not likely that the activity ratio V_ES_/V_EO_ of ROL is identical to the activity ratio of RML.

In summary, both enzymes prefer EO over ES as a substrate in the transesterification reaction with 1-propanol, but ROL shows a more obvious preference towards EO than RML. The enzyme dosage did not seem to influence the activity ratio when using either lipase. This verified difference motivated further structural studies to pinpoint potential differences in enzyme/substrates interactions.

### 2.2. Lipases Sequence Validation

Prior to more detailed structural studies, the sequence of each enzyme was subjected to peptide mass fingerprinting, PMF, to validate relevant residues in the respective enzyme preparation as compared to deposited sequences (in Genbank) and structures (in PDB). The PMF results showed that the peptide fragments obtained from commercial ROL (DF15) resulted in a 100% sequence identity and 60% query coverage of the peptide fragments generated in silico from the ROL sequence P61872 using the software MASCOT (Figure 2). The deposited sequence P61872 is in turn identical to the sequence of the deposited structure in PDB (1LGY), which corresponds to the closed lid structure of ROL (no open-lid structure is however available). The peptide fragments obtained from RML resulted in a 100% sequence identity and 42% query coverage of the in silico generated fragments of the deposited sequence P19515 (Figure 2), for which the open lid conformation is available in PDB (4TGL). The identified peptides covered most of the relevant residues proposed to be included in the substrate binding site of relevance for the binding of oleate and stearate (see also below). The result report of mass spectrometry (MS)/MS analysis is attached in Appendix A. As the 3D-structure of the open lid conformation was not available for ROL, a model was needed to allow further comparison of the substrate interacting residues in the two enzymes.

### 2.3. Homology Modeling of Open-Lid Structure of ROL

Most lipases are active at lipid–water interfaces, a function that is enabled by a mobile lid domain located over the active site, which controls the conditions for lipase catalytic activity. In pure aqueous media, the lid is predominantly closed, whereas in the presence of a hydrophobic layer, it is partially opened, allowing substrate interaction to occur. To study the substrate interactions in a lipase, an enzyme with an open lid conformation is subsequently necessary. As no crystal structure of ROL in the open lid conformation was available in PDB, a homology model of open-lid ROL (ROLop) was generated from the sequence data in PDB 1LGY (the structure of the closed lid ROL) using the RML open-lid structure (PDB 4TGL) as a template (57.74% identity and 99% coverage). The sequences of 4TGL and ROLop are aligned in Figure 3. Following homology modeling, energy minimization was applied to the model in YASARA, and the homology model was validated with MolProbity on the SWISS-MODEL server, which showed the QMEAN value of the model was −1.47 and the MolProbity score was 2.09 (which is a log-weighted combination of the clashscore, percentage Ramachandran not favored and percentage bad side-chain rotamers [19]). As the MolProbity score of the model was lower than the resolution of the template 4TGL (2.60 Å), the model was considered to be of good quality, as a numerically lower MolProbity score than its actual crystallographic resolution is, from an energy and stereochemical interaction point of view, better than the average structure at that resolution [19]. By superposing the generated model with the open-lid RML crystal structure 4TGL (Figure 4a) and closed-lid ROL crystal structure 1LGY (Figure 4b), it was also shown that the lid region of the generated open-lid ROL model is well aligned with the lid region of the crystal structure of the open-lid RML despite the differences in sequence, whereas the remaining part of the ROL structure was not affected. The model of the open lid structure of ROL, now allowed docking of the oleate and stearate and structural comparison of substrate interacting residues in the two targeted enzymes.

### 2.4. Structural Features of ROL and RML Related to Substrate Interactions

Stearate and oleate ligands were, respectively covalently docked into 4TGL and ROLop to mimic a possible covalent intermediate complex during the transesterification reaction. The docking complexes which held the highest binding energy in every simulation were selected as the most probable covalent complexes and refined with *md_refine* macro in YASARA [20]. These complexes are named RMOL, RMST, ROOL, and ROST. The binding energy of every complex is shown in Table 2a. The residues in the respective enzyme that could hydrophobically interact (within a distance of 4 Å) with either of the ligands (OL or ST) in every complex are illustrated in Figure 5.

As mentioned in the introduction, a long crevice region on the molecule surface, covered by the lid when the lipase molecule is not activated, is highly relevant for the substrate binding of ROL and RML (Figure 4c). In the four docking complexes which were finally chosen as the primary candidates based on binding energy, all showed the fatty acid ligand docked in this crevice, which is well aligned with results from previous research [21]. This also confirmed the validity of the complex models generated in this study. The substrate binding region of ROL and RML are structurally conserved with many of the relevant amino acid residues being identical, and several others being similar from a functional perspective (Table 3). An exception is the residue Trp88 in RML, which is an alanine (Ala89) at the corresponding location in ROL (Figure 4c).

The amino acid residues that were identified to be able to form interactions with the ligands in the docking complexes are well aligned with those proposed in previous research, which is supporting the validity of the docked complexes [18]. In both RMST and ROST, the oxyanion hole residues, where the backbone nitrogen can form hydrogen bonds with the carbonyl oxygen on the fatty acid ligand to stabilize the enzyme-ligand intermediate, are proposed to be Ser82(Thr83) and Leu145(146) [21,22]. The location of the structurally identical amino acid residues is numbered based on their location in the sequence of RML and ROL (in brackets) in this work. An interesting finding is that in RMOL complexes, the previously reported oxyanion hole residue Leu145 did not hydrogen bond the nitrogen with the ligand. Instead, the ligand formed a hydrogen bond with the hydroxyl group on the side chain of Tyr28(28) as shown in Appendix A. The distance between the carbonyl oxygen on oleate and the hydroxyl oxygen of Tyr28 in ROOL (4.20 Å) is also rather similar compared with the distance between the carbonyl oxygen and nitrogen on the backbone of Leu146 (3.76 Å), which may form potential hydrogen bonds. This indicates that Tyr28(28) plays an important role in anchoring the enzyme-ligand intermediate. The Tyr28(28) has been reported to be an important residue for catalysis by being an anchor group between the catalytic serine Ser144(145) and the lid region It has also been suggested that removal of the hydroxyl group of Tyr28 resulted in a decreased stabilization of the tetrahedral substrate intermediate based on free energy calculations [22], which is comparable to the finding in this study.

### 2.5. Differences in Interactions That May Explain the Higher Relative Activity of RML V_ES_/V_EO_ over ROL

From the activity profiles of the transesterification reactions, it could be seen that both RML and ROL showed a preference towards EO over ES as substrate (V_ES_/V_EO_ < 1) (Table 1), which however was less pronounced for RML. This preference was corroborated when comparing the binding energy of the docking complexes. For both ROL and RML, the complex with docked oleate has a higher binding energy, which indicates a more stable interaction between the ligand and the enzyme (according to YASARA binding energy-calculation), compared with the complex with stearate. The amino acid residues involved in hydrophobic interactions between the stearate and oleate also differed somewhat in the respective docking complex (Table 3), as explained in more detail below.

For RMOL and RMST, most of the residues included in the ligand interaction were identical, except Leu92 which only interacted with the oleate ligand in the RMOL complex. The type of interaction also differed for Tyr28 and Leu 145 (Table 3). Comparing ROOL and ROST, there are four residues that were not included in the substrate interaction in both complexes. In the ROST complex the ligand interacted with Phe214 and Ile254, which was not observed for ROOL, whereas in the ROOL complex the ligand interacted with Ile 93, and hydrogen bonded with Tyr28, which was not observed for the ROST complex (Table 3 and Figure 5). These differences may be explained by the difference in flexibility between oleate and stearate, where the cis double bond between C9 and C10 in oleate adds rigidity to the ligand in a way that increases the fit in the active site crevice, thereby increasing the binding energy.

In ROST, the docking results show that a part of the stearate ligand was less deeply bound in the crevice (from C17 next to the carboxyl carbon which is covalently bond with the catalytic Ser145) (Figure 5b). The docking pattern of ROST, together with the lower binding energy, indicates that the interaction between the stearate ligand and ROL is not as stable as in RML, which is also shown by the lower V_ES_/V_EO_ value of ROL (Table 1). As mentioned previously, many of the amino acid residue positions included in the substrate interaction in the four docking complexes, were, however, identical (Table 3), whereas some residues were not identical but with similar side chains, such as Ser/Thr or Leu/Ile. An exception is Trp88 in RML, which is Ala89 on the identical position in ROL. The bulky structure of Trp88 is highly likely to influence the substrate interaction in RML. The influence of Trp88 on RML and in general the four docking complexes were further validated with 50 ns MD simulations.

### 2.6. MD Simulation

The stability of the four generated docking complexes was investigated with 50 ns molecular dynamic simulation with cyclohexane as solvent to mimic the hydrophobic environment where normally lipases are applied. Ligand movement root-mean-square deviation (RMSD) and conformation RMSD of four docking complexes are shown in Figure 6. Root mean square fluctuation (RMSF) for each residue in the four complexes are presented in Appendix A, which shows no fluctuation larger than 1.5 Å in any of the complexes.

The stearate showed a more significant movement in some periods of the MD simulation (blue curves in Figure 6a,c), despite during most of the simulation the stearate docking complex stayed relatively stable in the substrate binding region. Compared with stearate, the movement of the oleate ligand was neglectable (orange curves in Figure 6a,c). However, the oleate ligand complexes showed a more obvious variation in conformation RMSD compared with stearate complexes (Figure 6b,d), although the variation (2 Å) is much less significant compared with the variation of the movement RMSD (6 Å) of the stearate complexes (Figure 6a,c).

Figure 7 superposed the input starting complexes with the representative snapshot at the time point when the movement RMSD (for stearate complexes) or conformation RMSD (for oleate complexes) was at the peak of the amplitude. From the MD simulation result of RMST and ROST, it is shown that stearate kept a stable conformation but experienced significant movement out from the substrate binding region in both complexes. On the other hand, Figure 7 also shows that the part of oleate ligand between the covalent bond to the enzyme and the double bond (C_18_ to C_9_) stayed stably at same the position in the substrate binding crevice during the 50 ns time period. The higher conformation RMSD (Figure 6b,d) is here caused by the movement of the region between C_1_ and C_9_ which moved similar to a “tail” during the simulation. For stearate, however, which is lacking the structural rigidity caused by the double bond, the whole ligand moved (Figure 6a,c) without significant conformational change (Figure 6b,d) at the period showing high-movement RMSD. This difference between the ligands, could be due to that the double bond between C_9_ and C_10_ reduced the structural flexibility of oleate that subsequently formed a stronger interaction with the benzene ring on Phe95 (96) which anchors part of the oleate (from C_18_ to C_10_) inside the substrate binding crevice.

Comparing RMST and ROST, the ligand movement RMSD in RML is less than in ROL, which is concluded to be due to the presence of Trp88 in RML as shown in Figure 7. The bulky sidechain of Trp88, compared with the Ala89 at the identical location in ROL, stabilized the interaction between RML and the stearate ligand during the MD simulation. In addition, an in silico mutagenesis simulation, showed that Trp88 could stabilize the ligand in the complex with decreased binding energy in W88A in RML and increased binding energy in A89W in ROL (Table 2b).

## 3. Materials and Methods

### 3.1. Material

Lipase ROL (powder) originating from *Rhizopus oryzae* was a gift sample from Amano Ltd. (the lipase has the trade name DF15). Lipase RML (liquid) originating from *Rhizomucor miehei* was manufactured by Novozymes (the lipase has the trade name RM) and purchased from Sigma-Aldrich Sweden AB.

The microporous polypropylene powder used for enzyme immobilization was under the trademark MP-1000 (used name as EP-100) from Akzo Nobel. The particle size selected was under 500 μm. p-Nitrophenol and Bradford reagents used to evaluate immobilization were purchased from Sigma-Aldrich Sweden AB.

Ethyl oleate (EO), ethyl stearate (ES), and 1-propanol were used as substrates in transesterification reactions and ethyl laurate (EL) used as an internal standard for gas chromatography (GC) analysis were purchased from Sigma-Aldrich Sweden AB. Propyl oleate and propyl stearate used as standards to calibrate product concentration in GC analysis were purchased from Larodan AB (Solna, Sweden). All the other chemicals used in product analysis were purchased from Sigma-Aldrich Sweden AB.

### 3.2. Lipase Immobilization

Both ROL (Lipase DF15; Amano Ltd., Nagoya, Japan) and RML (Novozym RM; Novozymes, Bagsværd, Denmark) were immobilized since the transesterification reaction is performed in a non-aqueous system. ROL and RML were immobilized on micro polypropylene carrier MP-1000 with hydrophobic interaction with the method described by Šinkuniene and Adlercreutz [8]. The immobilization yield of protein and activity were evaluated spectrophotometrically using the Bradford protein assay (595 nm) and p-nitrophenol butyrate lipase activity assay (405 nm), respectively [23].

### 3.3. Transesterification Reaction

Transesterification reactions with premixed EO:ES (molar ratio 1:1) and 1-propanol were done with a series of loadings of immobilized ROL and immobilized RML, respectively. The molar ratio between fatty acid esters and 1-propanol was 3:1, which gave a final molar ratio of 1.5:1.5:1 between the substrates. The reactions were run with different enzyme loading (shown in Table 4), with each reaction run in triplicate. In each reaction, immobilized lipases were mixed with 70.1 μL of 1-propanol and 1ml of 1:1 mixed EO-ES. The reactions were performed in 4 mL vials with MHR 13 Hearing-ThermoMixer (Hettich, AB Ninolab, Municipality in Sweden, Sweden) at 60 °C, under 900 rpm for 10 min with sampling at 1, 2, 3, 5, 7, 10 min. A 5 μL sample at each time point was mixed with 495 μL cyclohexane containing 25 mM EL and analyzed with GC.

### 3.4. GC Analysis

The 5 μL of sample from each reaction was diluted as described above and injected in a Thermo Scientific TRACE1300/1310 GC, with an FID front detector and SSL front inlet (Thermo Fisher Scientific S.p.A, Milan, Italy). The components were separated on a Supelco^®^ NukolTM column (15 m × 0.25 mm, 0.25 µm film) from Sigma-Aldrich Sweden AB (Stockholm, Sweden). Helium was used as a carrier gas. The temperature of the detector was set to 300 °C. Products, propyl-oleate (PO) and propyl-stearate (PS), were quantified from the relative peaks area and response factors calculated from standard curves made with PO and PS standards.

### 3.5. Lipase Sequence Validation

For bioinformatic simulation purpose, Peptide Mass Fingerprinting (PMF) [24] was performed on both commercial ROL and RML using ESI-Orbitrap MS/MS and MASCOT [25] to validate if the amino acid sequences of the commercial lipases are aligned with the sequences from Swiss-Prot https://www.uniprot.org/uniprot/, accessed on 12 May 2021) used for in silico simulation.

Both ROL and RML are crude mixtures, and they contain many impurities besides the lipases. To increase the purity of the commercial lipase so that it can meet the requirement of running MS/MS, SDS-PAGE was performed to separate the impurities from the lipase product. The gel fragment with the correct molecular weight was cut and digested with trypsin in-gel [26]. After the in-gel digestion, lipases were digested into constituting peptide fragments and their mass was determined by MS/MS analysis. The peptides of ROL and RML showed up as different peaks and the peak list was compared with the peak list generated from the in silico digestion of the ROL (Uniprot ID: P61872) and RML (Uniprot ID: P19515) collected from Swiss-Prot.

### 3.6. Homology Modeling

The crystal structures of both closed-lid RML (PDB ID: 3TGL) [27] and open-lid RML (PDB ID: 4TGLl) [28] available in the RCSB PDB Database (https://www.rcsb.org/, accessed on 15 June 2021) [29] have been used as templates. The crystal structure of closed-lid ROL (PDB ID: 1lgy) [30] has been published and the homology modeling of the open-lid structure of ROL was carried out with SWISS-MODEL server homology modeling pipeline (https://swissmodel.expasy.org/, accessed on 15 June 2021) [31]. 4TGL was chosen to be used as the template to simulate the open-lid structure of ROL since it is the first-rate template that holds an open lid ranked by SWISS-MODEL server searched from BLAST [32] and HHblits [33] database. The quality of the homology model was validated with MolProbity [19] on the SWISS-MODEL server.

### 3.7. Substrate Docking

Docking was performed using AutoDock implemented in the YASARA program [34], using the default docking parameters supplied with AutoDock in the ‘examples’ subdirectory. The point charges were initially assigned according to the AMBER03 force field [35], and then damped to mimic the less polar Gasteiger charges used to optimize the AutoDock scoring function. To simulate a possible covalent intermediate between the lipase molecule and ES or EO during the enzymatic reaction, a stearate and an oleate ligand were, respectively covalently docked into both an open-lid RML molecule, and an open-lid ROL molecule, using the script *dock_runcoval* macro in the YASARA program [34]. The structure of stearate and oleate (as PDB files) were built in Chimera 1.14 from their SMILES string searched from PubChem [36] and then energy minimized with the Molecular Modelling Toolkit (MMTK) which is plugged in Chimera 1.14 [37]. A covalent bond was made between the carboxyl carbon on the fatty acid ligands and the oxygen on the R-chain of the catalytic serine (OG atom, Ser144 for RML, Ser145 for ROL) on the lipase molecules. In each docking, both the lipase molecule (receptor) and fatty acid molecule (ligand) were set as flexible structures, and 25 docking runs of the covalently bound ligand were clustered to 2 distinct complex conformations which differs by at least 5.0 Å heavy atom RMSD after superposing on the receptor [34]. The one which had a more positive binding energy was chosen as the more preferred docking complex and then further refined with *md_refine* macro built in YASARA [35].

### 3.8. Molecular Dynamic Simulation

The MD simulations were run with the internal YASARA *md_run* macro, using cyclohexane with the density of 0.779 g/mL as the solvent to mimic the hydrophobic environment in the experimental reactions. Each simulation was run for 50 ns with a sodium chloride concentration of 0.9% to neutralize the simulation cell and a pH of 7.4. Before the simulation, steepest descent and simulated annealing minimizations were done to remove clashes. During the simulation, AMBER14 force field was used for the solute, GAFF2 and AM1BCC was used for the ligands. The motions equations were integrated with time steps of 1.25 fs and 2.5 fs for bonded and non-bonded interactions, respectively. The simulation cell was set as cubic with a cell extension of 10 Å on each side of the protein and a periodic boundary.

## 4. Conclusions

In this research study, we investigated the structural features in two lipases, ROL and RML from the filamentous fungi superfamily, that could influence their substrate preference between oleic acid and stearic acid esters (shown as different relative activities). From the experimental results, it is shown that both RML and ROL showed higher specific activity towards oleic acid esters than stearic acid esters and this difference was more significant for ROL than RML. A substrate docking simulation following by 50 ns MD simulations with cyclohexane as solvent elucidated that stearate had less stable molecular interactions with the substrate binding crevice in both RML and ROL with a higher tendency to move out of the substrate binding region, compared with oleate whose structure is more rigid due to the existence of the double bond. However, Trp88 in RML which is an Ala at the identical position on ROL significantly stabilized the substrate interaction in RML, which was especially evident with stearate as ligand. Phe95 located closely to the double bond of oleate also showed a possible stabilization effect in the substrate interaction with the C10 and C18 part of the oleate ligand.

## Figures and Tables

**Figure 1 ijms-23-07072-f001:**
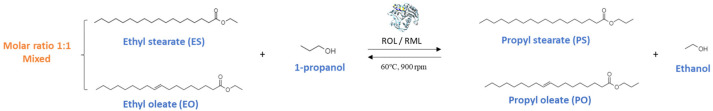
Transesterification reactions of ROL or RML with 1:1 molar mixed ES-EO and 1-propanol. The product PS and PO were quantified with gas chromatography (GC) and the relative activities V_EO_/V_ES_ of ROL and RML were, respectively calculated and compared.

**Figure 2 ijms-23-07072-f002:**
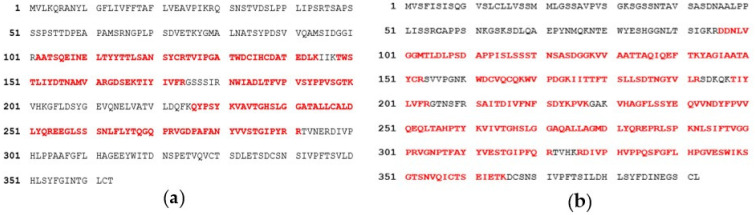
MS/MS analysis of lipases RML and ROL. Matched peptides between digested lipases samples and in silico digestions of lipases sequence from Swiss-Prot are shown in red, whereas the remaining part of the complete sequence is shown in black. (**a**) RML sample and RML sequence P19515; (**b**) ROL sample and ROL sequence P61872.

**Figure 3 ijms-23-07072-f003:**
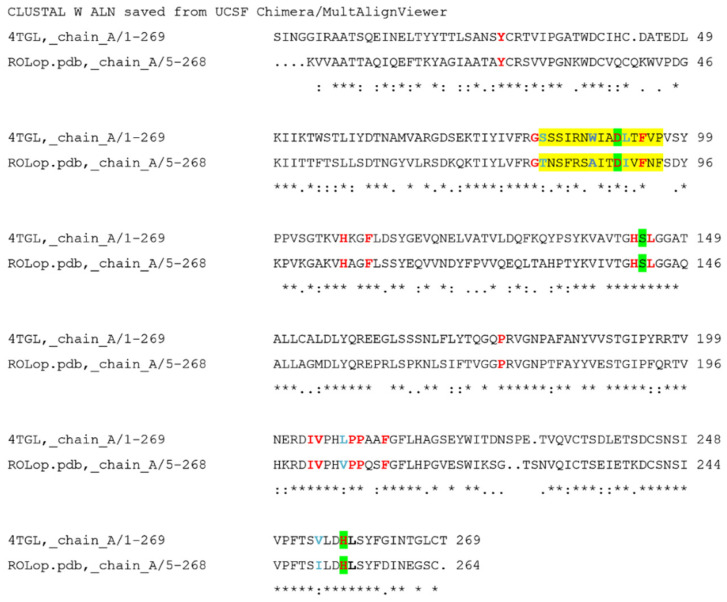
Structure based sequence alignment of 4TGL (RML) and ROLop. The catalytic triad residues are highlighted in green and the residues constructing the lid region are highlighted in yellow. Residues marked in red are located in the active site, i.e., potentially substrate interacting, and are identical in both sequences. Residues marked in blue are potential substrate interacting residues, at corresponding positions in the two structures, but are not completely conserved. Most of the residues, interacting with RML and ROL are displaying similar chemical properties (red and blue), except W88 in RML which at the identical position is A89 in ROL (also marked in blue).

**Figure 4 ijms-23-07072-f004:**
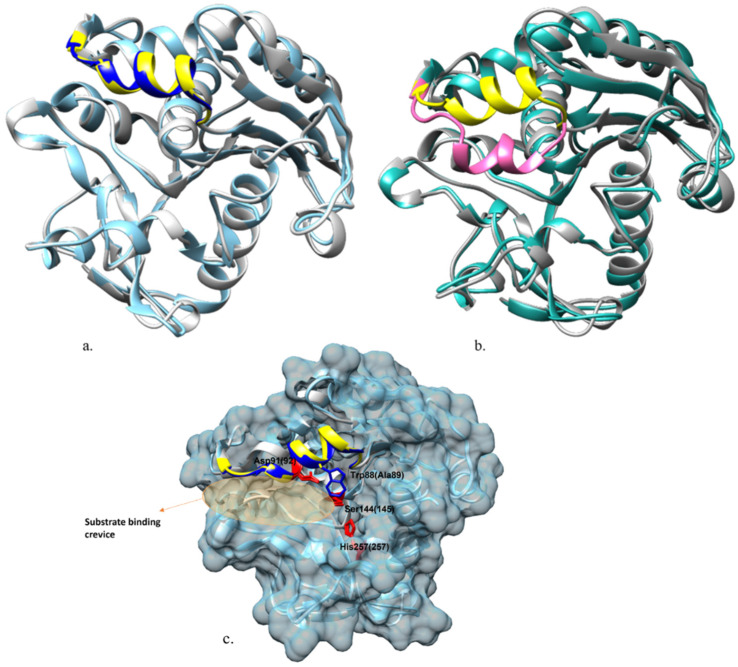
Superposing ROLop (grey) on relevant crystal structures: (**a**). Open-lid RML crystal structure 4TGL (ice blue). Yellow: lid region of the homology model; blue: lid region of 4TGL. (**b**). Closed lid ROL crystal structure 1LGY (cyan). Yellow: lid region of the homology model; hot pink: lid region of 1LGY. (**c**). (**b**) Catalytic essential characteristics identified including the catalytic triad shown in red and substrate binding crevice specified.

**Figure 5 ijms-23-07072-f005:**
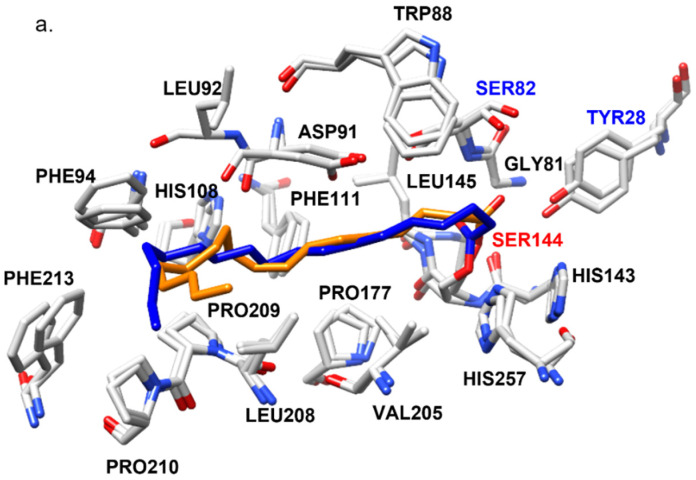
Docking complexes comparison with superposition. (**a**). Superimposing RMOL (amino acid residues marked in dark grey and the oleic acid ligand marked in orange) and RMST (amino acid residues shown in light grey and stearic acid ligand shown in bright blue). (**b**). Superimposing ROOL (amino acid residues shown in gold and the oleic acid ligand shown in orange) and ROST (amino acid residues shown in yellow and stearic acid ligand shown in bright blue).

**Figure 6 ijms-23-07072-f006:**
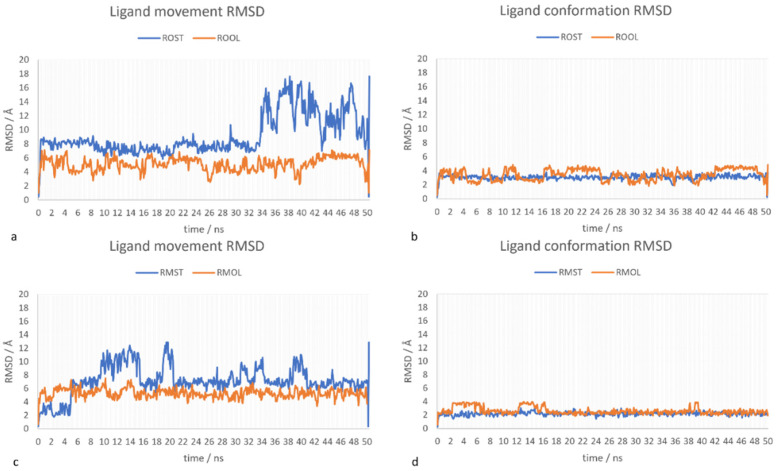
Ligand RMSD after superposing on the ligand as a function of simulation time generated from 50 ns MD simulation with cyclohexane as solvent. Plot (**a**). Ligand movement RMSD of ROST and ROOL; (**b**). ligand conformation RMSD of ROST and ROOL; (**c**). ligand movement RMSD of RMST and RMOL; (**d**). ligand conformation RMSD of RMST and RMOL.

**Figure 7 ijms-23-07072-f007:**
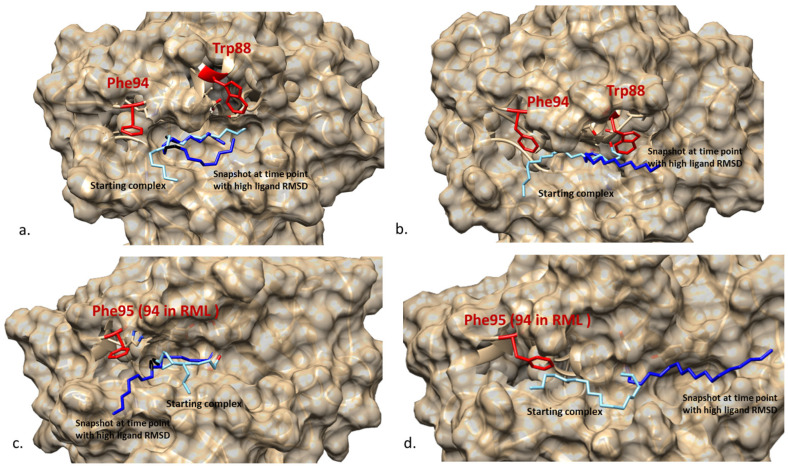
Superposition of the “fixed” starting complexes (ice blue ligand) of MD simulation and the snapshot (bright blue ligand) generated at the time point where the amplitude of the variation of either ligand movement RMSD (stearic acid) or ligand conformation RMSD (oleic acid) is the highest. (**a**). RMOL (snapshot at 14 ns); (**b**). RMST (snapshot at 13 ns); (**c**). ROOL (snapshot at 24.5 ns); (**d**). ROST (snapshot at 40 ns). Phe94 (Phe95 for ROL complex) and Trp88 which shows limitation on the ligand movement are marked as red on the figure.

**Table 1 ijms-23-07072-t001:** Activity profiles of ROL and RML in transesterifications.

Enzyme	Enzyme Dosage/mg	Activity towards EO: V_EO_/(mM/min)	Activity towards ES: V_ES_/(mM/min)	Relative Activity: V_ES_/V_EO_ *
Immobilized ROL	50	16.5 ± 1.5	10.8 ± 1.0	0.65
	60	21.2 ± 3.7	13.2 ± 2.6	0.62
	70	24.3 ± 1.2	15.1 ± 0.6	0.62
Immobilized RML	10	9.8 ± 1.9	7.2 ± 1.4	0.73
	20	20.9 ± 0.8	15.9 ± 0.7	0.76
	30	28.4 ± 1.5	21.7 ± 0.9	0.76

* The relative activity data are used for further statistical analysis.

**Table 2 ijms-23-07072-t002:** (**a**). Binding energy of docking complexes. (**b**). Binding energy of docking complexes.

(**a**)	
Docking complex	Binding energy/kcal/mol
Oleic acid docked in RML (RMOL)	8.43
Stearic acid docked in RML (RMST)	7.86
Oleic acid docked in ROL (ROOL)	6.52
Stearic acid docked in ROL (ROST)	5.63
(**b**)	
Docking complex	Binding energy/kcal/mol
Oleic acid docked in RMLW88A	6.07
Stearic acid docked in RMLW88A	5.70
Oleic acid docked in ROLA89W	7,81
Stearic acid docked in ROLA89W	6.10

**Table 3 ijms-23-07072-t003:** Amino acid residues interacting with the ligand in docking complexes. The green cell means that the residue can potentially form a hydrogen bond with the ligand. The orange cell means that the residue formed a hydrophobic interaction with the ligand. The blank cell means that the residue did not interact with the ligand. The residues of ROL are shown followed by the residues of RML in brackets.

	ROOL	ROST	RMOL	RMST
Tyr28(28)				
Gly81(82)				
Ser82(Thr83)				
Trp88(Ala89)				
Asp91(92)				
Leu92(Ile93)				
Phe94(95)				
His108(109)				
Phe111(112)				
His143(144)				
Leu145(146)				
Pro177(178)				
Ile204(205)				
Val205(206)				
Leu208(Val209)				
Pro209(210)				
Pro210(211)				
Phe213(214)				
Val254(Ile254)				
His257(257)				

**Table 4 ijms-23-07072-t004:** Lipases and loading dosage in transesterification reactions.

Reaction No.	Lipase Name	Loading Dosage (mg)
1	Immobilized ROL	50
2	Immobilized ROL	60
3	Immobilized ROL	70
4	Immobilized RML	10
5	Immobilized RML	20
6	Immobilized RML	30

## Data Availability

The data generated in this study are available in the published article.

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
