# Peer review of "Investigation of Structural Features of Two Related Lipases and the Impact on Fatty Acid Specificity in Vegetable Fats"

_ijms, 2022, doi:10.3390/ijms23137072_

Round 1

Reviewer 1 Report

The authors of the present manuscript perform an attempt to decipher the structural characteristics that confer substrate specificities to two lipases. In specific they examine the transesterification efficiency of a Rhizomucor miehei lipase (RML) and Rhizopus oryzae lipase (ROL) against oleic and stearic ethyl esters and then perform homology modelling, ligand docking and molecular dynamic simulations in order to structurally explain their findings. I believe that they should produce the RML W88A and A89W ROL mutants in order to biochemically prove their hypotheses.

Additional comments:

1.     Regarding the lipases activity profile, I do not see why the authors included both EO and ES in the reaction mixtures. Since the lipases prefer one of the two esters and the authors measure initial velocity, I think it would be better to do the transesterification reactions separately for each ester, and then compare initial rates.

2.     Regarding the modeling of ROL open conformation, since its sequence is known and the model was based on RML open-lid structure (PDB 4TGL), it is not clear what the authors mean in the following statement “As the open-lid crystal structure of ROL was missing, a homology model of open-lid 154 ROL (ROLop )was successfully generated from the sequence data in PDB 1LGY…”

3.     Figure 3: the authors should clarify whether this is a structure based sequence alignment. Legend: not all identical residues are shown in red. Also it is not clear what the authors mean by “structural identical”. Regarding the modeling of ROL open conformation, since its sequence is known and the model was based on RML open-lid structure (PDB 4TGL), it is not clear what the authors mean by “a homology model of open-lid ROL (ROLop )was successfully generated from the sequence data in PDB 1LGY “. Please explain.

4.      Figure 5: the aminoacid labels should be better positioned to be clear to the reader. There is no need to also include the chain (A).

5.     Lines 195-196: Figure 5 does not display the interactions.

6.     Lines 260-261: based on table 3, this is also the case for the other lipase and oleic acid (ROOL). The authors should state that. Also it is the backbone nitrogen that can form an H-bond, since leucine has no nitrogens on its side chain. This should be corrected in the following phrase as well.

7.     Reference no.9 is not correct

Author Response

We thank the reviewers for their valuable comments to our manuscript, and a point by point response is given below (in red font).

The authors of the present manuscript perform an attempt to decipher the structural characteristics that confer substrate specificities to two lipases. In specific they examine the transesterification efficiency of a Rhizomucor miehei lipase (RML) and Rhizopus oryzae lipase (ROL) against oleic and stearic ethyl esters and then perform homology modelling, ligand docking and molecular dynamic simulations in order to structurally explain their findings.

I believe that they should produce the RML W88A and A89W ROL mutants in order to biochemically prove their hypotheses.

Although the comment from the reviewer is very interesting, we believe that it is outside the scope of the current investigation, where we wanted to focus on the differences in activity ratio between two related lipases due to their structures (including the modelling of the open-lid form of ROL), thus allowing to link the activity profile to the structural features of the respective enzyme. For the latter part, the focus in the current work is the analysis of potential substrate interacting residues in the respective enzyme. We however, appreciate the reviewer´s suggestion, and it is our aim to include mutational studies in a future work, after the optimization of a production host organism.

  1. Regarding the lipases activity profile, I do not see why the authors included both EO and ES in the reaction mixtures. Since the lipases prefer one of the two esters and the authors measure initial velocity, I think it would be better to do the transesterification reactions separately for each ester, and then compare initial rates.

We discussed to either mix the two esters or separate them into two reactions, and we decided to mix them, due to the following reasons: i) this type of relative activity, will give a good indication of the competition between the two substrates, ii) we think that mixing can reduce any type of systematic or human errors introduced by pipetting, sampling and different rounds of reactions and analysis. We have put in an explanation in the manuscript to clarify this choice. (page 2 lines 94-96)

  1. Regarding the modeling of ROL open conformation, since its sequence is known and the model was based on RML open-lid structure (PDB 4TGL), it is not clear what the authors mean in the following statement “As the open-lid crystal structure of ROL was missing, a homology model of open-lid 154 ROL (ROLop) was successfully generated from the sequence data in PDB 1LGY…”

Generally, lipases are active at lipid–water interfaces, a function that is enabled by a mobile lid domain located over the active site, which controls the conditions for lipase catalytic activity. In pure aqueous media, the lid is predominantly closed, whereas in the presence of a hydrophobic layer, it is partially opened, allowing substrate interaction to occur. To study the substrate interactions in a lipase, an enzyme with an open lid conformation is subsequently necessary. As no crystal structure of ROL in the open lid conformation was available in PDB, this conformation was modelled. 1LGY is the PDB number of the crystal structure of ROL with a closed lid, which is matching the part of the enzyme that is not involved in the lid movement. However, since the lid has to be open in order for substrate interaction to occur (in the open lid conformation, the active site is exposed to, and can interact with the substrate molecules), a model of ROL with an open lid (ROLop) is necessary to allow studies of substrate interaction via docking simulations. As no crystal structure of open-lid ROL was available in PDB, this conformation was created by homology modelling. Subsequently, in this research study, ROLop was generated with the sequence information (fasta file) provided by 1LGY (the sequence corresponding to the lid part in the closed lid structure) and the crystal structure of RML in open-lid conformation (4TGL) as models. The resulting homology model corresponds to an “open-lid” structure of ROL that can be used in comparisons with the corresponding open lid structure of RML. We have added an explanation in the text of this section to further clarify the purpose of the modelling. (p. 4, lines 156-162)

  1. Figure 3: the authors should clarify whether this is a structure-based sequence alignment. Legend: not all identical residues are shown in red. Also it is not clear what the authors mean by “structural identical”. Regarding the modeling of ROL open conformation, since its sequence is known and the model was based on RML open-lid structure (PDB 4TGL), it is not clear what the authors mean by “a homology model of open-lid ROL (ROLop )was successfully generated from the sequence data in PDB 1LGY “. Please explain.

       Apologies for the unclear description.

Yes, the sequence alignment is structure based. The alignment was generated by superposing the ROLop sequence and the 4TGL sequence with Chimera. We have clarified this in the figure legend to Fig. 3 (line 182)

Legend: With the term “structural identical” we meant that the referred amino acid residues share an identical location in the active site on the respective structure. If the amino acid residues are conserved both structurally and chemically, they are marked in red.  If the amino acid residues located at the identical location on the respective structures are not completely conserved but share similar chemical structure on the side chain (e.g., Val vs Leu), the residues are instead shown in blue. We have rephrased the legend to make this more clear and added this explanation in the figure legend to Fig. 3, on lines 182-190.

      Concerning the modelling of open-lid ROL: as stated in the answer to Q2 above, the lid has to be open in order for substrate interaction to occur, which means that in the open lid conformation, the active site is exposed to and interacts with the substrate molecules. As no crystal structure of open-lid ROL was available in the PDB database, this conformation had to be created by homology modelling. The model was subsequently validated (with MolProbity on the SWISS-MODEL server, showing a QMEAN value of -1.47 and a MolProbity score of 2.09, as indicated in section 2.3, showing that the model is of good quality, and the model was thus considered successfully generated. We have now rephrased the explanation in section 2.3 (p 4-5), to clarify what was done and how this was evaluated.

  1. Figure 5: the amino acid labels should be better positioned to be clear to the reader. There is no need to also include the chain (A).

We agree. The marking of chain a is now removed, and the position of the label is improved.

  1. Lines 195-196: Figure 5 does not display the interactions.

The sentence is corrected to state that it is the potentially interacting residues (with a distance of max 4Å) that are shown.

  1. Lines 260-261: based on table 3, this is also the case for the other lipase and oleic acid (ROOL). The authors should state that. Also it is the backbone nitrogen that can form an H-bond, since leucine has no nitrogens on its side chain. This should be corrected in the following phrase as well.

We thank the reviewer for observing this. Compared with RMOL, ROOL has not only potential possibility to make hydrogen bond with Tyr28, but also with Leu146. This is later stated in the text. (Lines 277-280). We have corrected the Table 3 to include the potential Leu146 hydrogen bond (in ROOL) and corrected the text to state “backbone nitrogen” in the following phrase.

  1. Reference no.9 is not correct

      We thank the reviewer for noticing this, and have corrected the reference.

Reviewer 2 Report

The Manuscript submitted by Zehui Dong describes a study on two fungine lipases, which might be interesting in biotech.

The work is well done and described and it can be interesting for a wide audience.

Just two minor observations.

1 – References are sometimes inappropriate. For ex., line 45. Ref. 5 describes CATH and not the evolutionary relationship of a specific superfamily of proteins. And line 428, Ref. 26 does not describe the Protein Data Bank.

2 – Lines 162-165. In my opinion, the fact that a model is better than its template, in terms of MolProbity score, is absolutely irrelevant. Any model (homology, threading, AI) can be stereochemically perfect. Even a completely wrong model cab be stereochemically perfect. It is just a matter of refinement.

Author Response

We sincerely thank the reviewer for the nice comments. Please see the attachment for the detailed response.

Kind regards

Zehui

Round 2

Reviewer 1 Report

The authors have responded to my remarks and I believe the manuscript can be accepted for publication.

Author Response

Dear reviewer,

Thank you very much for your comments and approval!

Best regards,

Zehui